# Natural History of NAFLD

**DOI:** 10.3390/jcm10061161

**Published:** 2021-03-10

**Authors:** Raluca Pais, Thomas Maurel

**Affiliations:** Institut de Cardiométabolisme et Nutrition, Hôpital Pitié Salpetrière, Assistance Publique Hôpitaux de Paris, 75013 Paris, France; thomas.maurel@aphp.fr

**Keywords:** fatty liver, fibrosis, metabolic syndrome, insulin resistance

## Abstract

The epidemiology and the current burden of chronic liver disease are changing globally, with non-alcoholic fatty liver disease (NAFLD) becoming the most frequent cause of liver disease in close relationship with the global epidemics of obesity, type 2 diabetes and metabolic syndrome. The clinical phenotypes of NAFLD are very heterogeneous in relationship with multiple pathways involved in the disease progression. In the absence of a specific treatment for non-alcoholic steatohepatitis (NASH), it is important to understand the natural history of the disease, to identify and to optimize the control of factors that are involved in disease progression. In this paper we propose a critical analysis of factors that are involved in the progression of the liver damage and the occurrence of extra-hepatic complications (cardiovascular diseases, extra hepatic cancer) in patients with NAFLD. We also briefly discuss the impact of the heterogeneity of the clinical phenotype of NAFLD on the clinical practice globally and at the individual level.

## 1. Introduction

Closely related to the 21st century epidemic of obesity and type 2 diabetes, the prevalence of non-alcoholic fatty liver disease (NAFLD) continues to increase with almost 1 billion people being affected globally [1]. Of concern, it is expected that because of a longer exposure to metabolic risk factors—in pediatric NAFLD [2] and aging population [3]—the prevalence of advanced forms of NAFLD will increase significantly [4]. Contrary to other chronic liver diseases where the etiological agent is either modifiable (e.g., alcohol consumption) or accessible to a targeted specific treatment (viral or autoimmune hepatitis), multiple factors (genetic, epigenetic, environmental, clinical) are driving the disease progression in NAFLD, resulting in a variety of clinical phenotypes that require an individual therapeutic approach. For these reasons, it is particularly important to understand the natural history of NAFLD and identify the factors that can modify the disease course. Until now, our understanding of the natural history of NAFLD has come out from retrospective studies, either with paired liver biopsies or long-term follow-up, which are subject to heterogeneity and which make it difficult to adjust for multiple confounders. More recently, some prospective data began to emerge and give some new insight into the natural history of the disease. In this paper, we would like to critically analyze (1) the factors that drive the natural history of NAFLD, (2) the specific liver and metabolic clinical phenotypes and (3) their impact on the clinical practice and therapeutic advances globally and at the individual level.

## 2. Evolution of Histological Lesions

Thinking about the evolution of the histological lesions in NAFLD, we should refer to the spectrum of NAFLD and consider separately isolated steatosis or steatosis with minimal inflammation (non-alcoholic fatty liver, NAFL) and non-alcoholic steatohepatitis (NASH). Classically, it is considered that patients with NAFL do not progress or rarely progress to NASH and fibrosis. Experimental data have shown that triglyceride storage in the liver is not harmful per se and is rather an adaptive mechanism to increase free fatty acid influx, while lipotoxicity is responsible for the development and progression of hepatocellular injury, inflammation, hepatic stellate cell activation and extracellular matrix accumulation, which defines the phenotype of NASH [5]. However, clinical data from paired liver-biopsy studies have shown that around 25% of patients with NAFL progressed to NASH and bridging fibrosis. The presence of even mild inflammation on the baseline biopsy concomitant with the worsening of the metabolic risk factors could substantially increase the risk of progression when compared to isolated steatosis [6]. Among patients with NASH, the majority had either stable fibrosis (40%) or fibrosis progression (35%). The fibrosis progression rate is rather slow, of 1-stage over 14 years for patients with NAFL and 1-stage over 7 years in patients with NASH. However, important individual variations have been described, with some patients progressing faster than the others depending upon the presence and the diversity of the risk factors [7]. Although the fibrosis progression rate is rather slow, if we take into account the prevalence of pediatric NAFLD [8], then a sizeable proportion of patients will have advanced fibrosis or cirrhosis in their 50s years of age.

An important lesson from paired liver biopsy studies is that fibrosis progression is intimately linked to the evolution of the inflammatory lesions. Thus, a large study from the NASH-CRN registry has shown that an increase in the activity grade is associated with fibrosis progression and, conversely, a decrease in the activity grade and NASH resolution are associated with fibrosis regression [9]. The close association between changes in the fibrosis stage (either progression or regression) and the baseline activity scores or changes in the disease activity during follow-up is also seen in NASH clinical trials: patients achieving NASH resolution or a decrease in the activity grade also had some degree of fibrosis regression [10,11]. These results provide the rationale for developing drugs that targets disease activity and indirectly reverses fibrosis.

More recently, the evolution of the histological lesions in NAFLD has been described prospectively, based on the data derived from the placebo arms of clinical trials [12] or from real-life prospective follow-up cohorts [13]. Thus, in the simtuzumab clinical trial, the progression from bridging fibrosis to cirrhosis occurred in 22% of patients during a follow-up period of 29 months; fibrosis regression occurred in 21% of the patients with bridging fibrosis and in 9% of patients with cirrhosis. The proportion of patients with ≥1-stage fibrosis regression without worsening of NASH was even lower in phase 3 clinical trial with selonsertib, both in patients with bridging fibrosis (10%) or cirrhosis (13%) [14]. A recent meta-analysis of the placebo arms from clinical trials has shown that 30% of patients had the regression of the individual histological lesions (steatosis, inflammation and ballooning) and 21% had ≥1-stage fibrosis regression [12]. However, these data should be interpreted with caution as, because of the Hawthorne effect, it probably overestimates the spontaneous regression rates of the histological lesions in NAFLD seen in the real-life settings. For these reasons, prospective real-life cohorts are urgently needed.

Finally, when interpreting the histological data derived from clinical trials or real-life cohorts, we should be aware of several limitations related to liver biopsy. Liver biopsy only reflects the severity of the liver damage at a specific time point but does not capture the dynamic changes of the histological lesions. Change in the severity of the histological lesions is a nonlinear and dynamic process that is intimately related to the time of exposure and the evolution (improvement or worsening) of the metabolic risk factors.

## 3. Risk Factors for Fibrosis Progression

Identifying risk factors for disease progression is essential to allow a personalized approach for prevention strategies or pharmacological therapy (Figure 1). Contrary to viral hepatitis or alcoholic liver disease where the control of a single external factor—either viral replication or alcohol consumption—can induce fibrosis regression, the natural history of NAFLD is less predictable and much more sensitive to subtle variations of the associated risk factors. This concept might explain why the response rates in NASH clinical trials are much lower than those previously reported for other causes of chronic liver disease: 20% to 30% response rates in most of the phase IIb or III NASH clinical trials vs. > 90% sustained virological response reported in clinical trials with direct acting antiviral (DAAs) in patients with chronic hepatitis C. The severity of the liver damage in NAFLD is determined by various combinations of multiple risk factors and results in distinct clinical and histological phenotypes. The number and the severity of the metabolic abnormalities gradually increase across the spectrum of NAFLD from patients with NAFL to NASH without and with advanced fibrosis [15].

Although clustering of the metabolic risk factors is characteristic for NAFLD, the individual components of metabolic syndrome have different degrees of association with the severity of the disease. Natural history studies identified visceral adiposity and type 2 diabetes mellitus (T2DM) as the major drivers involved in diseases progression.

### 3.1. Visceral Obesity

It is now universally accepted that body fat distribution and visceral adiposity are more sensitive predictors for NAFLD severity than the body mass index (BMI). Compared to subcutaneous fat, the visceral adipose tissue is much more metabolically active and represents the major source of inflammatory cytokines, adipokines and free fatty acids [16,17]. A recent study of bariatric surgery patients has shown that the trunk/limb fat ratio and the adipocyte size progressively increased from the normal liver to steatosis and NASH/advanced fibrosis, and supports the hypothesis that fat storage shifts from subcutaneous tissue to other compartments such as visceral adipose tissue and the liver [18]. Both in lean and obese individuals, the expansion of the visceral adipose tissue, adipocyte hypertrophy and the accumulation of inflammatory macrophages are the main determinants of insulin sensitivity, increased level of inflammatory cytokines (i.e., interleukin 6 (IL6) and tumor necrosis factors alpha (TNFα)) [16,19,20,21] and accumulation of lipotoxic lipids (saturated fatty acids, free cholesterol, glycerolphospholipids and sphingolipids), ultimately responsible for the progression of the liver lesions [22]. Some studies have also shown a dose-dependent relationship between the amount of the visceral adipose tissue and the histological severity of NAFLD [23]. The role of visceral adipose tissue in the progression of NAFLD has been further emphasized in patients with lean NAFLD. These patients typically have low BMI but increased visceral adipose tissue and insulin resistance and, consequently, severe liver damage [24,25,26,27]. The individual histological lesions in NAFLD are very sensitive to absolute changes in body weight and body composition. Natural history follow-up studies have shown that absolute weight changes of ±5 kg from baseline tracks bidirectional changes in both activity and fibrosis scores [28]. Lifestyle interventional randomized clinical trials in NAFLD have shown a dose-dependent relationship between the magnitude of weight loss and histological changes: 5% weight loss is associated with regression of steatosis, while a more significant weight loss of ≥10% is associated with fibrosis regression [29].

### 3.2. Insulin Resistance and Type 2 Diabetes

Insulin resistance links T2DM and NAFLD through a bidirectional relationship and is responsible for both the occurrence of T2DM and for the progression of histological lesions in NAFLD. Euglycemic clamp studies have shown that excess visceral adipose tissue increases de novo gluconeogenesis, whereas liver fat is primarily associated with hepatic insulin resistance. Diabetic subjects with NAFLD have increased fasting insulin concentrations and are significantly more insulin-resistant than their diabetic counterparts without NAFLD [30].

NAFLD is present in almost 75% of patients with T2DM [31], whereas 25% of patients with NAFLD have T2DM [32]. Diabetic patients have almost 3- to 5-fold higher risk to be hospitalized or die because of NAFLD-related chronic liver disease [33,34]. The presence of NAFLD almost doubles the risk to develop new-onset T2DM [35] in close relationship with the severity of the histological lesions [36]. The concomitant presence of obesity and insulin resistance further increases the risk of new-onset T2DM (OR 3.23, 95% CI 1.78–5.89 and OR 14.13, 95% CI 8.99–22.2) [37]. Several Asian studies have shown that NASH resolution appears to diminish over time the risk of new-onset T2DM [38,39]. Longitudinal studies with paired liver biopsies have shown that the presence or the onset of T2DM during the follow-up is associated with fibrosis progression [6]. In bariatric surgery cohorts, the presence of advanced fibrosis is associated with lower remission rates of T2DM and, conversely, the presence and the severity of T2DM is a predictor for persistent fibrosis after bariatric surgery [40].

Compared to the general population, patients with T2DM had a significantly higher risk for all types of cancer, with the highest risk (hazard ratios, HRs) for liver (3.31), pancreas (2.19) and uterine cancer (1.78) [41]. Cohort studies from the US and Europe have shown that the risk of hepatocellular carcinoma (HCC) is increased 2- to 4-fold in diabetic patients without other risk factors [42,43] and is even higher in diabetic patients with concomitant alcohol consumption [44] or other causes of chronic liver diseases [45]. The HCC risk is greater in those patients with longer diabetes duration [46].

As a consequence of the impact of T2DM on NAFLD progression/complications, it is now recommended to screen for NAFLD in patients with T2DM and vice versa [47]. Large nutritional studies demonstrated a positive relationship between total sugar intake and cancer risk (HR for HCC, 1.88, 95% CI 1.16–3.03), suggesting that sugar intake may be a modifiable risk factor for cancer prevention overall [48,49]. Whether, controlling for T2DM, sugar intake and glycemic index might result in slower fibrosis progression rate and lower HCC risk, deserves further investigation (Figure 2).

### 3.3. Alcohol Consumption

It is widely recognized that alcohol consumption is a risk factor for severe liver damage, irrespective of the underlying cause of chronic liver disease. By definition, alcohol consumption in patients with NAFLD is low and does rarely exceed 20 g/day in women and 30 g/day in men [50]. Whether patients should be advised complete alcohol abstinence or if lower alcohol consumption below the above-mentioned thresholds should be accepted, is controversial. Most of the studies performed to date do not allow to assess for causality because of their cross-sectional design; these studies are very heterogeneous in terms of the type of data collected (i.e., the amount and the type of alcohol consumption, the drinking pattern, etc.). Several studies suggested that, compared to abstainers, patients with low-moderate alcohol intake are at lower risk for steatosis and fibrosis [51], but these studies did not completely adjust for confounders (i.e., physical activity, dietary factors, socio-economic status or metabolic comorbidities) or used surrogate endpoints which limits their conclusions [52]. Other studies suggested that, compared to lifetime abstainers, NAFLD patients with low alcohol consumption have significantly increased risk of HCC [53], cirrhosis decompensation (hazard ratio 1.7) or death/liver transplantation (hazard ratio 2.3) [54]. The deleterious effect of alcohol in patients with NAFLD is further accentuated by the interaction with the metabolic risk factors, in particular obesity (the risk fraction for liver related outcomes attributable to the interaction between the amount of alcohol and obesity varies between 25% and 70%) [55]. The pattern of drinking significantly impacts the future risk of liver damage—higher drinking frequency protects from steatosis while binge drinking is positively associated with significant liver damage [56]. Some studies also suggested that the type of alcoholic beverages differently impacts the severity of the liver damage: consuming 2 drinks/day of non-wine beverage doubles the risk for advanced liver disease compared to lifetime abstainers [55,57]. Most of the studies report a J-shaped relationship between the amount of alcohol and overall mortality, with 20% decrease in mortality risk in patients with <10g/day alcohol consumption compared with abstainers. These benefits are mainly due to reduction in cardiovascular mortality [57]. However, drinking ≥1.5 drinks/day was associated with worse outcomes in most of the studies [55,58].

### 3.4. Environmental and Genetic Factors

While obesity, T2DM or alcohol consumption can be modified through specific interventions, other factors like ethnicity or genetic risk factors are not modifiable (Figure 1). It is now widely accepted that NAFLD results from the complex interplay between clinical risk factors (particularly obesity and T2DM) and genetic variations. Multiethnic cohort studies highlighted a major inter-ethnic susceptibility for NAFLD independent of confounders—higher in Hispanics and lower in Africans [59]. The risk of severe liver damage is 12.5-fold higher in first-degree relatives of patients with NAFLD-related cirrhosis compared to general population [60]. GWAS studies identified robust and reproducible gene candidates including *patatin-like phospholipase domain-containing 3* (*PNPLA3*) [61,62], the *transmembrane 6 superfamily member 2* (*TM6SF2*) [59,63] and more recently the *17-beta hydroxysteroid dehydrogenase 13* (*HSD17B13*) [64,65], which are associated with the severity spectrum of NAFLD (from steatosis to NASH and advanced cirrhosis), HCC [66] or CV risk [67]. While single genetic variants are unable to provide an individual risk profiling, the development of polygenic risk score appears to be a more attractive approach that has been already validated in other pathologies (i.e., CV risk or breast cancer prediction) [68]. The specific interaction between clinical, environmental and genetic risk factors results in a unique individual profile with distinct pathophysiology translating into a specific clinical phenotype, risk of progression and response to treatment (Figure 1) [69].

## 4. Cryptogenic, NASH-Related Cirrhosis, End Stage Liver Disease (ESLD) and Liver Transplantation (LT)

Liver cirrhosis develops in 15% to 30% of patients with NAFLD usually at older age compared with other causes of chronic liver disease. This is probably due to a slower fibrosis progression rate compared to other chronic liver disease. However, even at a slow fibrosis progression rate, because of the prevalence of NAFLD in general population, it is projected that the prevalence of compensated and decompensated cirrhosis will significantly increase worldwide in the next decade [4]. Data from phase III clinical trials showed that in the absence of effective therapeutic interventions, 22% of patients with bridging fibrosis progressed to cirrhosis during 2 years of follow-up [70]. The risk of progression to cirrhosis increases with the clustering of the metabolic risk factors and doubles in patients with four metabolic traits [71]. The natural history of NASH cirrhosis is not different from other causes of chronic liver diseases. In compensated stages (Child A), patients with NASH cirrhosis had lower overall and liver-related mortality, but in decompensated stages (Child B and C cirrhosis), the prognosis is similar to other etiologies of chronic liver diseases [72].

Sometimes, in cirrhotic stages, the diagnosis of NASH is difficult to establish because of the absence of the typical histological lesions (some patients are labeled as “cryptogenic cirrhosis”) and requires either the documentation of NASH on a historical liver biopsy or past exposure to metabolic risk factors. Cryptogenic cirrhosis is a diagnosis of exclusion, in the absence of identifiable causes of chronic liver disease and it has been long time confounded with “burn-out” NASH—cirrhosis without the histological hallmarks of NASH. Because of the increasing awareness for NASH, data from both United Network for Organ Transplantation (UNOS) [73] and European Registry for Liver Transplantation (ELTR) [74] database show that, starting in the year 2000, the number of cases listed with the diagnosis of cryptogenic cirrhosis decreased while the number of patients listed with NASH-related cirrhosis increased. However, even nowadays, some patients are still classified as cryptogenic cirrhosis, suggesting that beyond the semantics, cryptogenic and NASH cirrhosis are distinct entities. Recent studies have shown that cryptogenic and NASH cirrhosis have different clinical features and prognosis. Patients with NASH cirrhosis are more often females, of older age and Caucasian origin and have higher prevalence of obesity and T2DM; they are more often listed for HCC and have more severe disease [75]. Patients with cryptogenic cirrhosis have evidences of more active fibrosis with greater collagen content and α-smooth muscle actin expression on liver biopsy and they develop liver-related events in a shorter period of time as compared with NASH-related cirrhosis [76].

NAFLD is now the second leading cause and the most rapidly growing indication for liver transplantation (LT) for both decompensated cirrhosis and HCC [77,78]. Remarkably, during the past 10 years, the prevalence of NAFLD as an indication for LT has increased by 170% [78]. This is explained by several factors: (1) a better awareness for the disease, (2) the epidemic of obesity and T2DM, (3) aging population which results in a (4) longer exposure to risk factors, and finally, (5) by a decrease of other causes of CLD, particularly viral hepatitis C. Patients with NAFLD on the waiting list for LT are older, have higher BMI and cardio-metabolic comorbidities and are considered at higher risk for LT. While on the waiting list, patients with NAFLD have higher MELD and are more likely to be removed from the list because of uncontrolled comorbidities. After LT, although the risk of recurrent NAFLD exists because of the persistence/worsening of metabolic risk factors [79], overall, the outcomes of patients with NAFLD are similar with those of patients transplanted for other etiologies [80,81]. In this context, coupled with the gap between the number of indication for LT and the availability of the liver grafts, the selection of patients with NAFLD that will mostly benefit from liver transplantation is challenging and requires a multidisciplinary team for optimal evaluation and control of metabolic and cardiovascular risk factors. Although the guidelines for evaluation of patients with NAFLD candidates for LT begin to slowly evolve [82], each center should establish the accepted risk tolerance on individual basis.

## 5. Hepatocellular Carcinoma

Although HBV and HCV are still the leading causes of HCC, their prevalence is expected to decline because of the vaccination policies in HBV and the development of direct acting antivirals (DAAs) allowing to cure HCV. Because of the epidemics of obesity and metabolic syndrome, the prevalence of NAFLD-HCC continues to rise, placing NAFLD as the leading cause of HCC globally. Among patients with NAFLD-related cirrhosis, the HCC incidence ranges from 2.4% to 12.8% [83]. In cirrhotic patients, the carcinogenic mechanisms are similar to other chronic liver diseases and follow the classical sequence of altered DNA methylation, specific mutational signature, structural genomic lesions and activation of different carcinogenic pathways [84]. The particularity of NAFLD-HCC is the occurrence in the absence of cirrhosis in 40% to 50% of patients [85], suggesting the involvement of additional oncogenic pathways in relationship with low-grade inflammation associated to obesity and metabolic syndrome [86] (Figure 3).

Obesity is associated with increased release of cytokines from dysfunctional adipose tissue, particularly TNF-α and IL-6, which further activates pro-oncogenic pathways involving NF-κβ, JNK and mTOR and controls cell proliferation and apoptosis. Hyperinsulinemia increases the circulating levels of insulin-like growth factor 1 (IGF1) and thus inhibits apoptosis and favors cell division [87]. These mechanisms explain the particularities of the clinical phenotype of HCC in NAFLD and a slower rate of oncogenic transformation in NAFLD. In line with that, patients with NAFLD-HCC are a decade older than patients with HCV-HCC and almost two decades older compared to HBV-HCC patients [88]. NAFLD-HCC is often diagnosed at a later stage, with more advanced disease because of the lack of surveillance strategies. Because of a delayed diagnosis, patients are less eligible for curative interventions [89] but for the same tumor stage, the prognosis of NAFLD-HCC is similar to HCC associated to other causes of chronic liver disease [90]. Because almost half of NAFLD-related HCC occurs in the absence of cirrhosis, it is a major challenge to develop risk stratification models and to adapt the surveillance strategies. Although the current guidelines recommend HCC screening only in patients with cirrhosis [91], several aspects should be considered. First, the “*non-cirrhotic*” stages covers a large spectrum of liver disease severity from the absence of fibrosis to bridging fibrosis, which is actually a “*precirrhotic*” stage; whether patients with bridging fibrosis should benefit from HCC screening is of debate. Second, when these guidelines have been developed, the recommendations targeted mainly patients with viral hepatitis. Although some “disease modifiers” (Figure 3), including clinical (i.e., presence and clustering of the metabolic traits) [71], biological (transaminases levels) [92] and genetic (PNPLA3, TM6SF2, HSD17B13) [93,94] factors allow for a “*general*” estimation of the HCC risk, these approaches are not accurate enough to justify the implementation of the screening strategies in the absence of cirrhosis. A polygenic approach [86], coupled with clinical risk factors, could potentially be useful to refine HCC risk in patients with non-cirrhotic NAFLD but its use in every day practice has to be validated.

## 6. Extra-Hepatic Complications of NAFLD

### 6.1. Cardiovascular Disease

Given the high prevalence of both NAFLD and cardiovascular (CV) diseases and their association with the metabolic syndrome [95,96], it is not surprising that these two entities frequently coexists and impact significantly the healthcare system. Cross-sectional studies have shown that NAFLD is associated with a broad spectrum of CV disease—from early atherosclerosis to clinically manifested CV events and deaths both in general population or in specific patients groups [97]. Thus, case control studies and several meta-analyses have shown that NAFLD is associated with increased carotid intima-media thickness [98], coronary artery calcifications [99], increased arterial wall stiffness [100], impaired endothelium dependent flow-mediated vasodilatation [101], early changes in left ventricular morphology and diastolic function [102], impaired myocardial energy metabolism [103] and coronary dysfunction [31]. NAFLD is also associated with increased risk of incident fatal and non-fatal CV events (OR 1.64; 95% CI 1.26–2.13) as shown by several cohort studies and a recent meta-analysis [104,105]. Long-term follow-up cohort studies have shown that, in the absence of cirrhosis, patients with NAFLD are more likely to experience CV-related death than liver-related death [54].

However, because of the cross-sectional, retrospective design and multiple concomitant confounders, these studies do not provide answers to several major questions: (1) the causal relationship between NAFLD and CV disease beyond the common risk factors; (2) whether NAFLD and the severity of liver damage in NAFLD additionally increase the CV risk; and (3) how the association between NAFLD and CV disease will impact on the clinical practice.

The causal relationship between NAFLD and CV disease is suggested by common physiopathological pathways beyond the insulin resistance and includes low-grade inflammation, endothelial dysfunction, increased oxidative stress and oxLDL, increased angiogenesis, hypercoagulability (increased PAI—1, increased coagulation factors VIII, IX, XI, XII, decreased protein C activity, etc.), altered gut microbiota, etc. [106,107]. Low-grade systemic inflammation (increased IL-6, IL-1β, TNF α, M1/M2 balance, etc.) induces endothelial dysfunction, alters endothelial tone and enhances plaque formation. Patients with NAFLD also have a more “atherogenic” lipid profile with decreased HDL and increased TG and LDL, particularly small dense LDL particle, which favors early atherosclerosis [22,108]. Gut dysbiosis related to NAFLD is associated with increased flavin containing monooxygenase (FMO) and trimethylamine N-oxyde (TMAO) and further impacts cholesterol metabolism and promotes foam cell formation and early atherosclerosis [109]. Clinical studies have confirmed a pro-coagulant imbalance in patients with NAFLD in close relationship with the severity of the liver damage [110].

In relationship with the inflammatory milieu described above, it is not surprising that the independent contribution of NAFLD to CV disease is strongest in more advanced disease settings, i.e., NASH and advanced fibrosis. Longitudinal studies have shown that NAFLD plays an active role in the progression of early atherosclerosis and indirectly support the causal relationship between NAFLD and CV disease [111,112]. Some studies have also shown that adding NAFLD to traditional CV risk factors improves their performance to predict CV events [113].

All together, these data support the recommendation of both the American and European Association for the Study for Liver Diseases (AASL and EASL) to carefully assess and control the CV risk in patients with NAFLD [50,114]. Although clinical studies have shown that NAFLD is present in 58% of patients requiring coronary angiogram and is associated with coronary artery stenosis and increased need for coronary intervention, these data are not strong enough to support screening for NAFLD in patients with CV disease [115].

Long-term data from phase III ongoing NAFLD clinical trials will answer the question of whether improving liver condition will have an impact on the occurrence of incident CV events.

### 6.2. Extra-Hepatic Cancer

The general knowledge is that concurrent obesity and type 2 diabetes patients with NAFLD are at increased risk of extra-hepatic cancer [31]. Long-term cohort studies have found extrahepatic cancer to be the second cause of death after CV disease [116,117]. Of the extrahepatic cancers, stomach, pancreas and colon have almost a 2-fold increase in incidence and a trend toward a younger age at diagnosis in NAFLD. A stratification of cancer risk by sex has also been observed: men with NAFLD were at higher risk for colon, liver and stomach cancer, while women with NAFLD are at higher risk for cancer of the liver, stomach and uterus [118]. It is under debate whether NAFLD plays an active role in carcinogenesis or is just an innocent by-stander associated with increased risk of malignancy in the presence of obesity and T2DM. Several studies have shown that BMI and particularly visceral obesity and weight gain are associated with increased risk of cancer [119,120,121]. However, other studies have shown that metabolically healthy obesity is not associated with increased risk of cancer, suggesting that additional mechanisms are involved in carcinogenesis [122]. These results have been confirmed by a recent study showing that NAFLD was associated with increased risk of cancer while obesity alone was not [123]. NAFLD has also been identified as an independent predictor of cancer risk in patients with T2DM [124]. Although these data allow drawing a clinical phenotype of NAFLD (based on age, sex, obesity and visceral adiposity, T2DM) at higher risk for specific type of cancer, screening for extrahepatic cancer in patients with NAFLD is not recommended.

### 6.3. Mortality and Costs

Several long-term follow-up studies have shown that patients with NAFLD have increased overall and specific mortality compared to the general population. However, most of these studies have significant biases related to the relatively small sample size, retrospective design, variable follow-up period (range from 1 year to 20 years in some studies), heterogeneity of the diagnostic criteria for NAFLD, and the small number of events, which limits the generalizability of the results [116,117]. A major question with a significant clinical impact on the monitoring and treatment policies is to determine at which point of the severity of the liver damage the mortality risk is beginning to rise. In other words, does the mortality risk increase in patients with NASH and mild/moderate fibrosis, in which case these patients should be monitored closely and proposed specific pharmacological interventions, or does the mortality risk significantly increase only in patients with advanced fibrosis, in which case only these patients should be targeted by specific therapeutic approaches? Most of the long-term follow-up studies have shown that fibrosis stage, but not NASH, is the only major and independent predictor for the clinical outcomes [116,117,125], but further studies have shown that the mortality risk increase from earliest fibrosis stages [126]. A recent population-based cohort study including 10,568 patients with histological proven NASH in Sweden, has shown that NAFLD was associated with 93% higher relative risk of overall mortality and a 20-year absolute excess risk of 15.3%. This risk increased in a dose-dependent manner with the spectrum of the histological severity in NAFLD: 10.7% higher in patients with simple steatosis, 18.5% higher in patients with NASH without fibrosis, 25% higher in patients with NASH and advanced fibrosis without cirrhosis and 49% higher in patients with NASH cirrhosis. In particular, when compared with patients with simple steatosis, patients with non-fibrotic NASH had an excess mortality rate of 5.1 per 1000 person-year, which means that, over 20 years of follow-up, 1 out of 10 persons with NASH without significant fibrosis will die [127]. The excess mortality risk in NAFLD is mainly related to extra-hepatic cancers and cirrhosis [54,118,128] as already emphasized in this review. Although the relationship between NAFLD and non-fatal CV events is widely accepted [97,129] and previous studies have suggested cardiovascular disease to be the major cause of death in patients with NAFLD, the specific contribution of NAFLD to excess cardiovascular mortality is being questioned by more recent studies and meta-analysis [104].

Because of its prevalence, its potential of progression and associated comorbidities, the impact of NAFLD on the healthcare system is significant. In 2016, it has been estimated that the total direct cost related to NASH was 103 billion $ in US and 28 billion € in EU-4 countries (Italy, Germany, France and Spain); these costs were even higher if considering the societal costs related to the disease [130]. Overall, the costs per patients were higher in the presence of advanced fibrosis (3983 €) than in the early stages (2224 €) [131]. Once patients have cirrhosis, the annual cost increased from the compensated to decompensated cirrhosis, HCC and liver transplantation [132]. The total cost of NAFLD over the next two decades in patients with T2DM it has been estimated at 1.67 trillion $. However, the liver-related costs, attributable at least to annual check-up, are estimated to be 13.7 billion $ in patients with simple steatosis and 163 billion $ in patients with NASH [133].

## 7. Conclusions

In conclusion, the natural history of NAFLD is very heterogeneous with some patients progressing faster than the others and covers a large spectrum of severity. It is now accepted that NAFLD is a multi-systemic disease situated in the center of the metabolic syndrome and insulin resistance, and involves not only the liver but also the extrahepatic organs. Therefore, the overall phenotype of NAFLD is determined by the severity of the liver damage (steatosis, NASH and different stages of fibrosis severity from F0 to F4, cirrhosis-related complications—HCC, decompensated cirrhosis), and by the number, the time to exposure and the control of the metabolic comorbidities. It is now accepted that the severity of the liver damage is determined by various combinations of multiple risk factors (clinical, genetic, environmental) and increases with the number of risk factors that are present. The heterogeneity of the disease is an important challenge in developing specific drugs for NAFLD and explains the failure of some of them. Therefore, in the absence of a specific therapy so far, it is essential to understand the natural history of the disease and identify those factors that predominantly drive disease progression in a given patient and that can be modified by specific interventions to curb the disease progression.

## Figures and Tables

**Figure 1 jcm-10-01161-f001:**
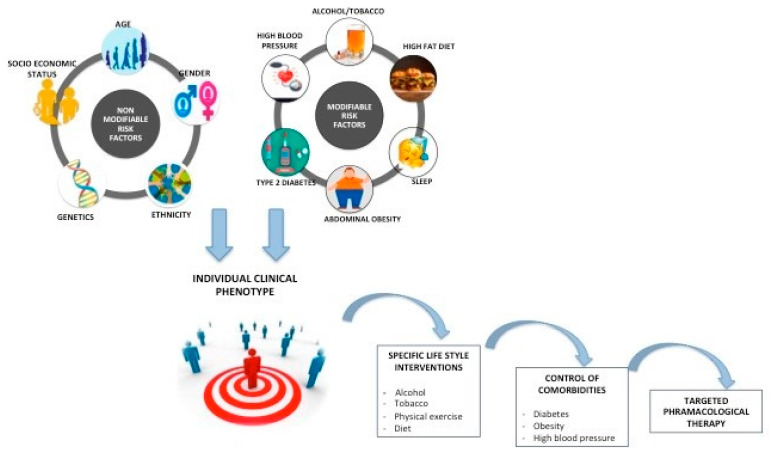
The severity of the liver damage in non-alcoholic fatty liver disease (NAFLD) is determined by the interaction of multiple risk factors combined in a well-determined clinical phenotype. Some of these factors (i.e., age, gender, genetics, etc.) cannot be modified, while others (obesity, type 2 diabetes, high blood pressure, sleep apnea) can be modified through specific interventions (general lifestyle measures or targeted control of comorbidities). Because of a unique clinical phenotype of each patient, NAFLD therapeutic interventions should be tailored on a clinical-based personalized approach for the selection of the best non-pharmacological approach or drug candidates to target a specific physiopathogenic pathway.

**Figure 2 jcm-10-01161-f002:**
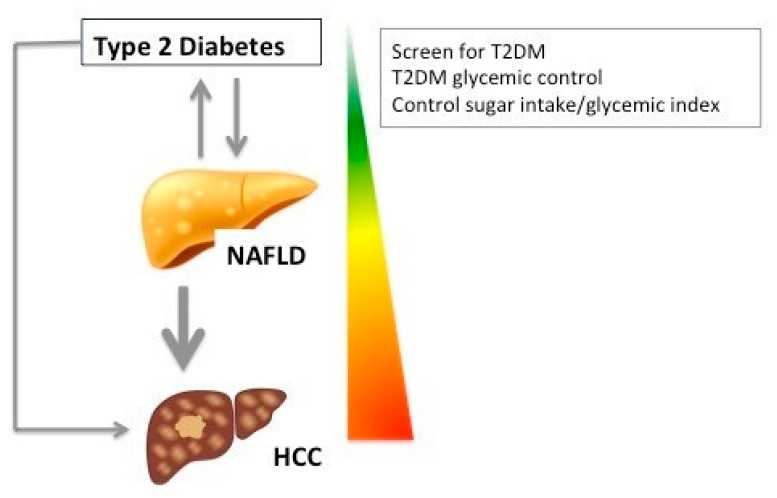
T2DM and NAFLD are linked through a bidirectional relationship: NAFLD increases the risk of T2DM occurrence and T2DM is a risk factor for NAFLD development and fibrosis progression. Total sugar intake and T2DM are independent risk factors for hepatocellular carcinoma (HCC).

**Figure 3 jcm-10-01161-f003:**
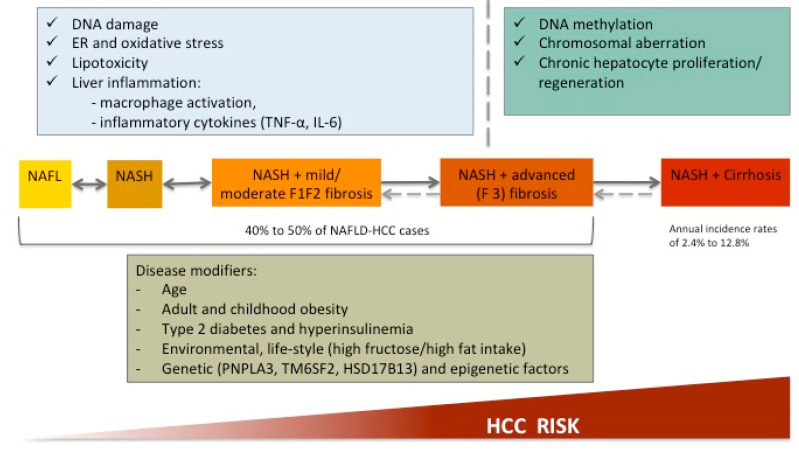
Mechanisms of carcinogenesis according to the presence/absence of cirrhosis and disease modifiers involved NAFLD-related HCC.

## Data Availability

Not applicable.

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
