# Peer review of "Natural History of NAFLD"

_jcm, 2021, doi:10.3390/jcm10061161_

Round 1

Reviewer 1 Report

no further comments

Reviewer 2 Report

The article has improved substantially and the authors have correctly reviewed all the questions raised.

This manuscript is a resubmission of an earlier submission. The following is a list of the peer review reports and author responses from that submission.

Round 1

Reviewer 1 Report

In this manuscript, the authors well summarized the current knowledge about the natural history of NAFLD/NASH. My minor comments follow:

  1. The authors should add a section about some new potential therapeutics for NASH\NAFLD under development.
  2. I could not read some characters in the figures. Please check you used widely recommended fonts in the figures. Or paste as images.
  3. Please check the style of references. Al least the authors have to revise the style of journal names.

Reviewer 2 Report

  1. Numerous abbreviations are not indicated the first time they are cited in the text. Authors should review the entire manuscript.
  2. Figures 1-3 should be self-explanatory, poorly understood, and too basic. Legends of the figures would be advisable showing the meaning of the images.
  3. The authors should review and clarify lines 41 and 47 as they seem contradictory.
  4. The expression and grammar of the entire text should be reviewed to narrate and abbreviate in some paragraphs.
  5. Line 133 IR ?, line 150 HRs ?, HCC should also be indicated the first time.
  6. p142. 05% should be 95%
  7. line 166. Missing reference.
  8. Numerous studies indicate that alcoholic cirrhosis is independent of the type of alcohol and pattern of consumption, only the amount of alcohol is important UBE, Legaz et al.2016 and Burra et al., 2019.
  9. The authors indicate several “GWAS studies” to refer to candidate genes, but they only name two genes and a single bibliographic reference.
  10. liver transplantation = LT
  11. The authors should be more concise, concluding different phenotypes that can be considered in preventive medicine. A summary table of all the aspects considered would be recommended.
  12. The manuscript is very scattered, it should be more precise and specific to the natural history of NAFLD. It ends with a fuzzy conclusion.